# Small molecule inhibition of Csk alters affinity recognition by T cells

Boryana N Manz[1†], Ying Xim Tan[1†‡], Adam H Courtney[1], Florentine Rutaganira[2,3], Ed Palmer[4], Kevan M Shokat[2,3], Arthur Weiss[1,3]*

[1]Rosalind Russell and Ephraim P. Engleman Rheumatology Research Center, Division of Rheumatology, Department of Medicine, University of California, San Francisco, San Francisco, United States; [2]Department of Cellular and Molecular Pharmacology, University of California, San Francisco, San Francisco, United States; [3]Howard Hughes Medical Institute, University of California, San Francisco, San Francisco, United States; [4]Departments of Biomedicine and Nephrology, University Hospital Basel, University of basel, Basel, Switzerland

*For correspondence: art.weiss@ucsf.edu

†These authors contributed equally to this work

Present address: ‡Division of Cancer Genetics and Therapeutics, Laboratory of Chromatin, Epigenetics and Differentiation, Institute of Molecular and Cell Biology, Singapore, Singapore

Competing interests: The authors declare that no competing interests exist.

**Abstract** The C-terminal Src kinase (Csk), the primary negative regulator of Src-family kinases (SFK), plays a crucial role in controlling basal and inducible receptor signaling. To investigate how Csk activity regulates T cell antigen receptor (TCR) signaling, we utilized a mouse expressing mutated Csk ($Csk^{AS}$) whose catalytic activity is specifically and rapidly inhibited by a small molecule. Inhibition of $Csk^{AS}$ during TCR stimulation led to stronger and more prolonged TCR signaling and to increased proliferation. Inhibition of $Csk^{AS}$ enhanced activation by weak but strictly cognate agonists. Titration of Csk inhibition revealed that a very small increase in SFK activity was sufficient to potentiate T cell responses to weak agonists. Csk plays an important role, not only in basal signaling, but also in setting the TCR signaling threshold and affinity recognition.

## Introduction

SFKs are ubiquitous regulators of basal and inducible receptor signaling. The seven family members are expressed in various combinations in different cell types, have unique substrate specificity and are differentially regulated by phosphatases and localization (*Lowell, 2004*). However, they share a common negative regulator Csk. How Csk maintains basal and inducible receptor signaling is still unclear.

In T cells, the strength and duration of T cell antigen receptor (TCR) signaling in response to antigen stimulation dictates the magnitude and quality of primary and secondary immune responses (*Smith-Garvin et al., 2009*; *Chakraborty and Weiss, 2014*). The TCR activation threshold, agonist affinity discrimination and signal termination must therefore be tightly regulated. This is achieved through the concerted action of multiple positive and negative regulators acting basally and during inducible signaling. The SFK Lck regulates TCR signaling by phosphorylating tyrosines in the cytoplasmic segments of the TCR ζ and CD3 chains, basally and during antigen recognition, as well as by phosphorylating downstream kinases such as ZAP-70 and ITK (*Chan et al., 1995*; *Heyeck et al., 1997*; *Smith-Garvin et al., 2009*). The availability and recruitment of active Lck has been proposed to be the rate-limiting step in discriminating agonist affinity (*Stepanek et al., 2014*). Weak agonists with shorter half-lives of pMHC (peptide-bound major histocompatibility complex)-TCR interaction have less time to recruit active CD4 or CD8 co-receptor bound Lck. Active Lck (phosphorylated on Y394) is critical for downstream signaling and can be detected in the basal state, but its level does not change appreciably after TCR stimulation (*Nika et al., 2010*).

**eLife digest** The immune system has 'T' cells that recognize when the body is infected with a virus or bacterium and mount an immune response that is targeted to that microbe. They can also find and eliminate cancer cells. The microbes and cancer cells produce molecules called antigens that are detected by proteins on the surface of the T cell called T cell receptors (TCR). Antigen recognition causes the TCRs to transmit signals to the inside of the T cell that trigger an immune response. The degree to which the TCRs are active, and the length of time that they transmit signals regulates the size of the immune response. Therefore, developing new drugs that manipulate the activity of TCRs could be useful to treat many diseases.

An enzyme called Csk inhibits the activities of a small family of proteins involved in a variety of different processes. One protein that Csk targets is called Lck and is required for the activation of immune responses in T cells. However, it is not clear whether Csk interacting with this protein stops the release of signals from TCRs, or whether it alters the level to which TCRs need to be activated before they transmit the signals.

Manz, Tan et al. studied T cells from mice that had a mutant form of Csk that is inhibited by a drug called 3-iodo-benzyl-PP1 (3-IB-PP1), but otherwise works normally. When these cells detected an antigen in the presence of the drug, its TCRs were more highly activated and transmitted signals for a longer period of time than cells not exposed to the drug. The drug especially enhanced immune responses to very weak antigens, ones that might not activate T cells under normal circumstances.

Manz, Tan et al.'s findings confirm that Csk plays a negative role in the activation of T cells and suggest that Csk may be a useful target for drug therapies that aim to fine-tune immune responses. The next challenge is to find a drug that can inhibit normal Csk enzymes and test this in mice and other animals.

Although the molecular regulation of Lck is understood, its localization and changes in activity during TCR signaling are still unclear (*Chakraborty and Weiss, 2014*). Lck is tightly regulated by phosphorylation on two conserved tyrosines. Trans-autophosphorylation of its kinase domain activation loop tyrosine, Y394, increases its catalytic activity, whereas phosphorylation of its C-terminal tail inhibitory tyrosine, Y505, promotes its closed, inactive conformation (*Palacios and Weiss, 2004*; *Chakraborty and Weiss, 2014*). In T cells, the receptor-like protein tyrosine phosphatase CD45 regulates Lck activity positively and negatively by dephosphorylating its inhibitory tyrosine and activation loop tyrosine (*McNeill et al., 2007*; *Zikherman et al., 2010*). However, the major negative regulator of Lck is Csk, which phosphorylates its inhibitory tail tyrosine (*Bergman et al., 1992*; *Levinson et al., 2008*).

Csk is a cytoplasmic protein, yet its predominant role is to phosphorylate the inhibitory tyrosine of SFKs that are localized to membranes (*Bergman et al., 1992*). Hence, multiple adaptors have been implicated in regulating its activity by influencing its localization. Transmembrane PAG and Lime, as well as the cytosolic PH-domain containing Dok1/2, may play roles in recruiting Csk to the cell membrane in a phosphorylation-dependent manner. The association of Csk with PAG in biochemically defined lipid rafts has been observed to diminish in response to TCR stimulation, leading to its proposed involvement in regulating TCR signaling (*Torgersen et al., 2001*; *Davidson et al., 2003*). Since Csk returns to these fractions upon signal termination, it has also been implicated in the down-regulation of signaling (*Torgersen et al., 2001*; *Davidson et al., 2003*). However individual genetic deficiencies in PAG, LIME or Dok1/2 have had little effect on TCR signaling and on T cell function (*Brdickova et al., 2003*; *Dobenecker et al., 2005*; *Xu et al., 2005*; *Yasuda et al., 2007*), indicating that our understanding of Csk regulation during TCR signal initiation, propagation and termination remains incomplete.

The precise role of Csk in TCR signaling has also been difficult to study by genetic inactivation (*Imamoto and Soriano, 1993*; *Nada et al., 1993*; *Schmedt and Tarakhovsky, 2001*). Germline deletion of Csk leads to embryonic lethality (*Imamoto and Soriano, 1993*; *Nada et al., 1993*). Conditional deletion of Csk in the T lineage leads to TCR-MHC-independent, but Lck-dependent, passage through both thymic beta selection and positive selection checkpoints, both of which require perception of pre-TCR or TCR signaling, respectively (*Schmedt and Tarakhovsky, 2001*). However, since Csk deficiency leads to aberrant T cell development and compensatory changes in

the basal state, studying the role of Csk in ligand-induced signaling in unmanipulated primary T cells has been hindered.

We recently generated BAC transgenic mice that express a mutant of Csk (Csk[AS]) in the absence of endogenous Csk (*Tan et al., 2014*). Csk[AS] kinase activity can be specifically and rapidly inhibited by an analog of the general kinase inhibitor PP1, 3-iodo-benzyl-PP1 (3-IB-PP1). Studies in thymocytes in this model system support the notion that the combined actions of CD45 and Csk establish the basal activity of SFK (*Tan et al., 2014*). However, it is not clear whether Csk plays a role in setting the TCR signaling threshold or in signal termination. In this study, we utilized CD4[+] and CD8[+] T cells from Csk[AS] mice to investigate how Csk modulates the threshold for TCR signal activation, affinity recognition and TCR signal termination. Our data suggest that by controlling the amount of available active Lck, Csk regulates the TCR signaling threshold and affinity recognition. Notably, even at a low level of Csk inhibition, T cell activation by weak agonists is greatly enhanced.

## Results

To determine whether Csk activity regulates the TCR activation threshold, we stimulated Csk[AS] CD4[+] T cells or wild type CD4[+] T cells (as a specificity control) by titrating anti-CD3ε antibody in the absence or presence of a high dose of the Csk[AS] inhibitor, 3-IB-PP1, and examined proximal TCR signaling tyrosine phosphorylation events. As previously shown, inhibition of Csk[AS] alone induced strong activation of Lck, as well as Fyn, which was indicated by the hyperphosphorylation of their activation loop tyrosines and the reduced phosphorylation of their inhibitory tyrosines (*Tan et al., 2014*). In contrast, ligation of the TCR alone did not detectably alter Lck phosphorylation, consistent with an earlier report (*Nika et al., 2010*). The striking Lck activation after Csk[AS] inhibition was associated with only relatively weak downstream phosphorylation of ZAP-70, LAT and PLC-γ1 (*Figure 1A*). However,

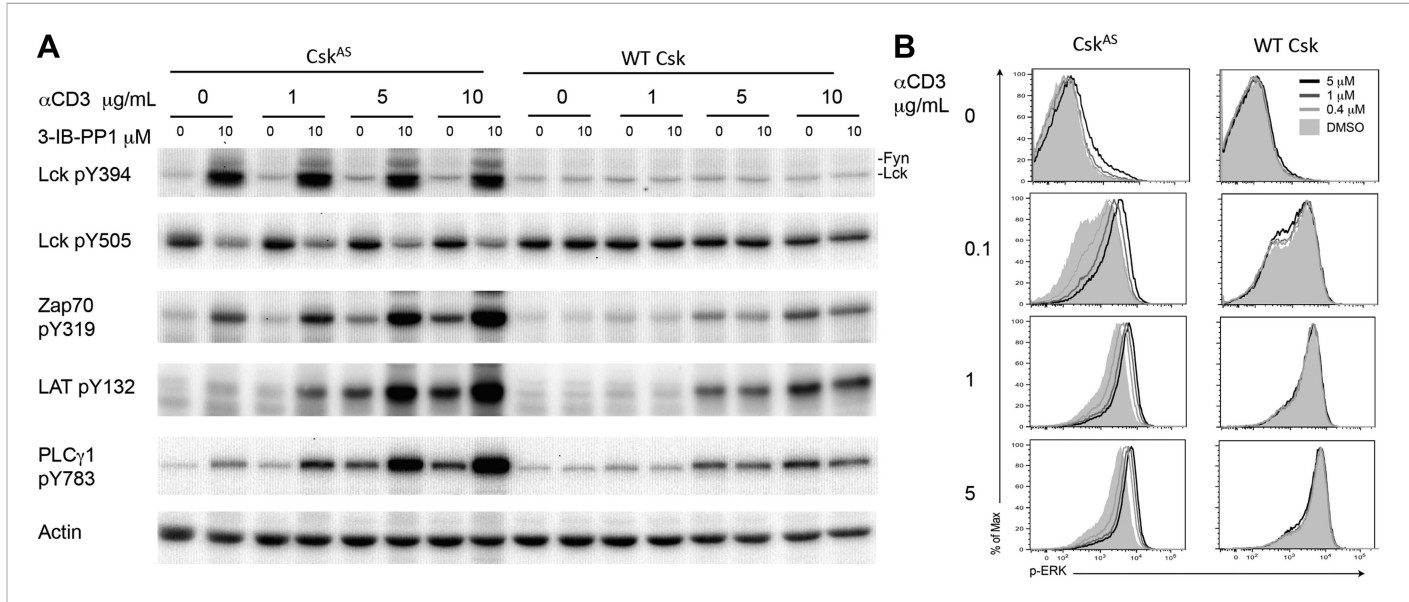

**Figure 1**. Inhibiting Csk increases the magnitude of ligand-induced TCR signaling and reduces the threshold for TCR activation. (**A**) Purified Csk[AS] (AS) or wildtype (WT) CD4[+] T cells stimulated for 2 min with 1 µg/ml, 5 µg/ml or 10 µg/ml anti-CD3ε antibody in the presence of DMSO or 10 µM 3-IB-PP1 were analyzed by immunoblotting for the phosphorylation of the activation loop tyrosine of Lck and Fyn (Src pY416 antibody) and the inhibitory tyrosine of Lck (Lck pY505), phosphorylated ZAP-70 (ZAP70 pY319), LAT (LAT pY132) and PLC-γ1 (PLCγ1 pY783), as well as total actin (loading control). Data are representative of at least three independent experiments. (**B**) Purified total Csk[AS] (AS) or wildtype (WT) T cells stimulated with the indicated dose of anti-CD3ε antibody for 2 min in the presence of DMSO or 5 µM, 1 µM or 0.4 µM 3-IB-PP1 were analyzed for phosphorylated ERK (p-ERK) by phosphoflow. Histograms were gated on CD4[+] cells. Data are representative of at least three independent experiments for AS cells and two independent experiments for WT cells.

The following figure supplement is available for figure 1:

**Figure supplement 1**. Inhibiting Csk during TCR stimulation prolongs TCR signals.

this downstream signaling was very strongly enhanced when the Csk$^{AS}$ inhibitor treatment was combined with anti-CD3ε crosslinking, as evidenced by the marked increase in ZAP-70, LAT and PLC-γ1 phosphorylation (*Figure 1A*). To determine whether this early enhanced proximal signaling was transmitted further downstream, we examined ERK1/2 phosphorylation (p-ERK). The magnitude of p-ERK was increased by Csk$^{AS}$ inhibition in a 3-IB-PP1 dose-dependent manner, with the greatest enhancement seen when using a low dose of anti-CD3ε (*Figure 1B*). At a high dose of anti-CD3ε the increased p-ERK signal intensity, as indicated by the slight peak shift for 3-IB-PP1 treated cells, might reflect a more rapid or more complete peak response at the time point assayed. Notably, Csk$^{AS}$ inhibition alone induced only a small amount of protein phosphorylation downstream of ZAP-70 (i.e., LAT and ERK1/2), likely due to spurious activation of a few cells (*Figure 1A,B*). Instead, Csk$^{AS}$ inhibition synergized with TCR stimulation leading to downstream signaling. Our data strongly suggest that Csk activity plays a crucial role in establishing the TCR activation threshold. Specifically, a reduction in Csk activity lowered the threshold for signal activation and increased TCR sensitivity.

It has been proposed that Csk plays an important role in TCR signal termination (*Torgersen et al., 2001*; *Davidson et al., 2003*). To test this model we examined a time-course of Csk$^{AS}$ CD4$^+$ T cells stimulation with anti-CD3ε antibody. Csk$^{AS}$ inhibition prolonged phosphorylation of ZAP-70, LAT and PLC-γ1 (*Figure 1—figure supplement 1A*). Moreover, there was still a clear digital response to anti-CD3 stimulation that was augmented by Csk$^{AS}$ inhibition at the earliest time point (*Figure 1—figure supplement 1B*). There was evidence of 3-IB-PP1 dose-dependent delay in downregulation of p-ERK, with the down-regulation seeming more heterogeneous and, perhaps, asynchronous compared to the digital up-regulation of the response (*Figure 1—figure supplement 1B*). However, Csk inhibition did not prevent eventual signal attenuation over time, indicating that Csk has only a partial role in signal termination. Other negative regulatory mechanisms must contribute to the termination of TCR signaling, albeit more slowly in the absence of Csk (*Naramura et al., 2002*; *Rhee and Veillette, 2012*).

To examine more physiologically relevant regulatory mechanisms induced by peptide-MHC stimulation, we generated Csk$^{AS}$ mice expressing the ovalbumin peptide/MHCII-reactive OTII TCR transgene (*Fu et al., 2013*; *Salmond et al., 2014*). A tetramer was used as the stimulus to allow for detailed biochemical analyses that would not have been possible with the confounding contribution of proteins from antigen presenting cells (APCs). Stimulation of Csk$^{AS}$;OTII CD4$^+$ T cells with an agonistic OVA pMHC tetramer during Csk$^{AS}$ inhibition resulted in markedly enhanced phosphorylation of ZAP-70, LAT and PLC-γ1 (*Figure 2A*) and led to increased and prolonged ERK1/2 phosphorylation (*Figure 2B*). Thus, our findings with the physiologic OVA pMHC TCR ligand parallel those using anti-CD3ε, confirming that Csk plays an important role in setting the TCR activation threshold and has a partial role in signal termination.

To assess whether the reduced TCR activation threshold for proximal signaling events following Csk inhibition had a functional impact on downstream T cell activation, we monitored cell proliferation of Csk$^{AS}$;OTII T cells stimulated with plate-bound anti-CD3ε (*Figure 2—figure supplement 1A*) or OVA pMHC tetramer (*Figure 2—figure supplement 1B*) in the presence of anti-CD28 costimulation. Each stimulus was titrated to doses that yield minimal to maximum responses since antibody and tetramer have different molecular weights and physiological potency. At low doses, both anti-CD3ε and OVA pMHC tetramer induced greater cellular proliferation when Csk$^{AS}$ was inhibited, consistent with the increased magnitude of proximal TCR signaling. Therefore, by controlling Csk activity, the threshold for TCR signaling and resultant responses can be modulated.

Next, we investigated how much perturbation of Csk and Lck is necessary to induce physiological changes in the T cell response. Here we used Csk$^{AS}$-expressing CD8$^+$ T cells or transgenic CD8$^+$ T cells with the OTI TCR specific for an OVA peptide/MHC class I complex. Titration of 3-IB-PP1 in Csk$^{AS}$ CD8$^+$ T cells revealed a maximum threefold to fourfold enhancement of Lck pY394, which correlated with increased Lck activity (*Figure 3A*) (*Chakraborty and Weiss, 2014*). Thus at resting state 25–33% of Lck was phosphorylated at Y394, which agrees with some previous estimates (*Nika et al., 2010*) and is higher than others (*Ballek et al., 2015*). It should be noted that this is only an inference from the plateau with high drug doses. This study is not designed to estimate the percentage of active Lck at the basal state, since some Lck may not be available for activation by Csk inhibition. Low doses of 3-IB-PP1 (1 μM or less) induced at most a 50% upregulation of pY394, while doses of 5–10 μM induced a threefold to fourfold upregulation. Surprisingly, phosphorylation of the inhibitory tail Y505, the direct target of Csk, did not decrease as appreciably. This observation suggests that Lck is actively

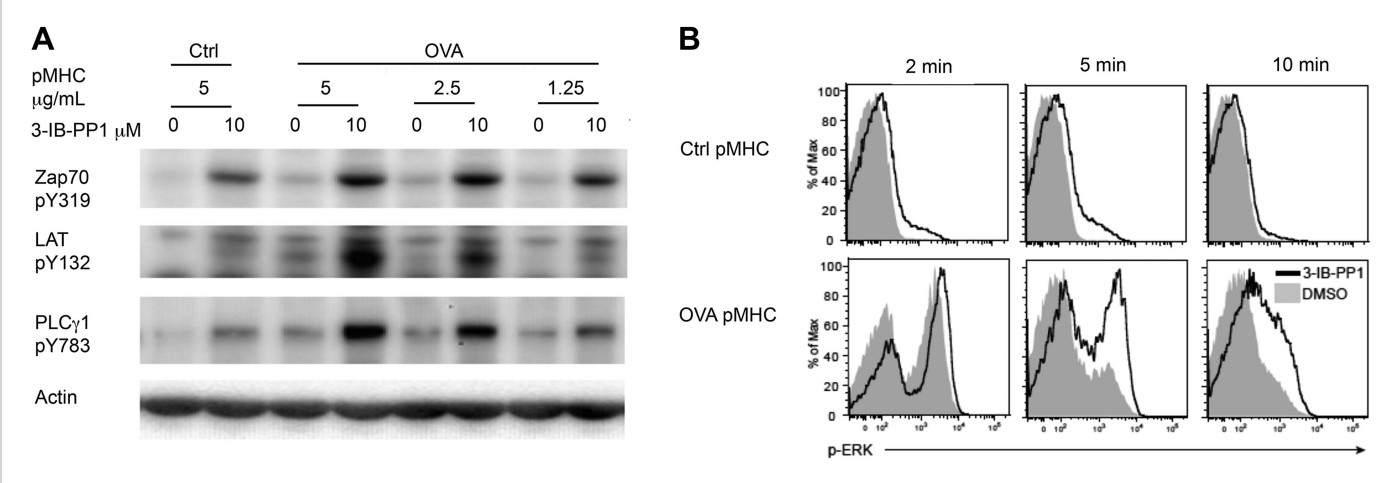

**Figure 2**. Inhibiting Csk reduces the threshold for TCR activation and prolongs signaling induced by pMHC engagement. (**A**) Purified Csk[AS];OTII CD4[+] T cells stimulated for 3 min with 5 µg/ml bead-bound control pMHC tetramer or 5 µg/ml, 2.5 µg/ml or 1.25 µg/ml bead-bound OVA pMHC tetramer in the presence of DMSO or 10 µM 3-IB-PP1 were analyzed by immunoblotting for the phosphorylated ZAP-70, LAT and PLC-γ1, as well as total actin (loading control). Data are representative of three independent experiments. (**B**) Purified Csk[AS];OTII CD4[+] T cells stimulated with 5 µg/ml bead-bound control-pMHC tetramer or 2.5 µg/ml bead-bound OVA pMHC tetramer for 2, 5 or 10 min in the presence of DMSO or 5 µM 3-IB-PP1 were analyzed for phosphorylated ERK (p-ERK) by phosphoflow. Histograms were gated on CD4[+] Vα2[+] cells. Data are representative of three independent experiments.

The following figure supplement is available for figure 2:

**Figure supplement 1**. Csk inhibition shifts the threshold for cellular proliferation in response to anti-CD3 or p-MHC tetramer stimulation.

phosphorylated by Csk, but a substantial proportion of pY505 is either inaccessible for dephosphorylation by CD45 or is dephosphorylated over a longer timescale. This, however, does not explain how the small reduction in pY505, particularly so in CD8[+] T cells, results in a large increase of pY394.

We explored whether the effects of Csk inhibition has differential effects on CD4[+] vs CD8[+] T cells. However, we found that CD4[+] and CD8[+] T cells have identical responses to Csk inhibition when normalized to their basal level of Lck phosphorylation (*Figure 3—figure supplement 1*). Interestingly, CD8[+] T cells express ~15% more Lck (*Olszowy et al., 1995*) and have a higher (~20%) basal amount of inhibitory tail pY505 phosphorylation. In both CD4[+] and CD8[+] T cells, pY394 increased much more markedly than the associated decrease in pY505 in response to low doses of the Csk inhibitor. The molecular mechanism for this response is unknown. One possibility to consider is that even a small pool of active Lck that is dephosphorylated at Y505 and becomes phosphorylated at Y394 enables a trans-autocatalytic mechanism which can become amplified within the larger pool of Lck molecules that are not phosphorylated on pY394. Interestingly, it has also been observed that stimulation with anti-TCR and anti-CD4 antibody leads to substantial increase in pY394 without much loss of pY505 (*Ballek et al., 2015*).

Short-term Csk[AS];OTI CD8[+] T cell stimulation with bead-bound OVA-MHC tetramer was enhanced even by low levels of Csk inhibition (<1 µM) (*Figure 3B*), as demonstrated by the degree of PLC-γ1 and ERK1/2 phosphorylation. At this inhibitor dose, Lck pY394 did not increase by more than 50%, suggesting that even subtle changes in Lck activity can potentiate proximal signaling. Higher doses of the inhibitor potentiated Lck signaling even further, suggesting Csk and Lck activity can be titrated over a wide range.

Lck not only initiates TCR-dependent signaling but is also proposed to be the crucial gate-keeper in kinetic proof-reading models that attempt to explain the exquisite affinity discrimination of T cells of agonists with only slightly different half-lives (*Palmer and Naeher, 2009*; *Stepanek et al., 2014*). We hypothesized that Lck hyperactivation may differentially regulate strong and weak agonists. To address this question, we utilized a panel of well-characterized altered OVA peptides (*Daniels et al., 2006*). When Csk[AS];OTI CD8[+] T cells were stimulated by APCs preloaded with peptides of different agonist potency during strong Csk[AS] inhibition (5 µM), we observed marked augmentation of CD69 expression in responses to weak (T4, Q4H7, G4) agonists, contrasting with only slight enhancement of

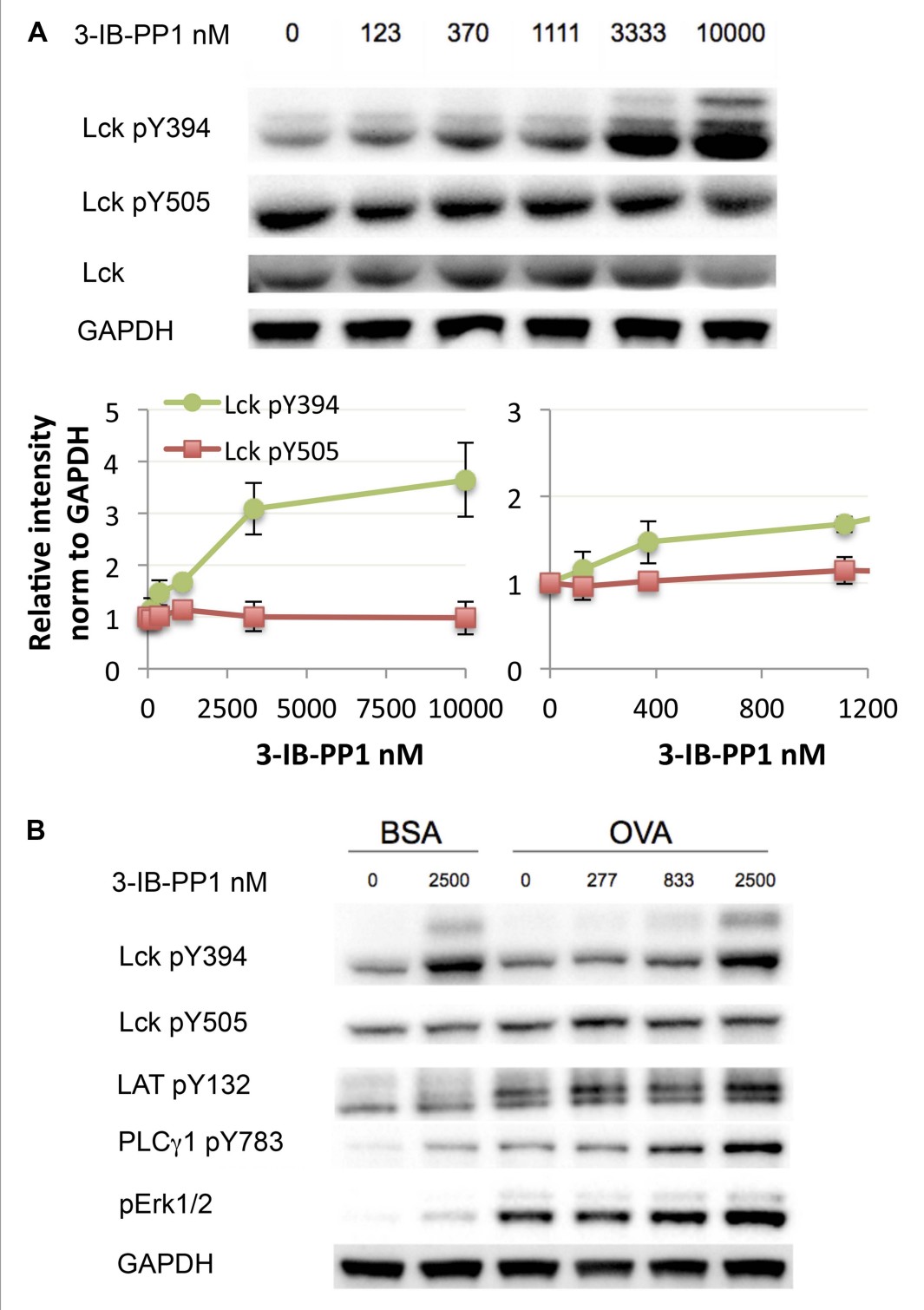

**Figure 3**. Weak Csk inhibition and Lck activation potentiate agonist signaling. (**A**) Csk[AS] CD8[+] T cells were rested and stimulated with indicated doses of 3-IB-PP1 for 3 min, lysed, and immunoblotted for activating (pY394) and inhibitory (pY505) site Lck phosphorylation. Below is quantification (± sem) of three independent experiments immunoblot. (**B**) Csk[AS];OTI naive CD8[+] T cells were rested, stimulated with bead-bound BSA or 10 μg/ml pMHC-OVA for 3 min, lysed and assayed by immunoblot. Data are representative of at least two independent experiments.
*Figure 3. continued on next page*

*Figure 3. Continued*

The following figure supplement is available for figure 3:

**Figure supplement 1**. Comparison of Csk inhibition and Lck activation in CD4[+] vs CD8[+] T cells.

the responses to strong (OVA, Q4R7) agonists (*Figure 4A*). Although the OVA response was not fully saturated at this early time point (4 hr), it was only slightly enhanced by strong Csk inhibition. Upon stimulation with OVA, but not altered peptides, the TCR is quickly downregulated due to its phosphorylation and retention in intracellular vesicles (*Cai et al., 1997*). Csk inhibition alone induced some ligand-independent TCR downregulation, but additional downregulation was observed with all peptides, even the weakest G4, which argues for altered signaling of low-affinity peptides at a very upstream step (*Figure 4—figure supplement 1*). Importantly, the null agonist VSV-loaded APCs did not induce similar activation or TCR downregulation, demonstrating that Csk inhibition works strictly in synergy with a cognate TCR ligand. Moreover, although the spleen-derived APCs should still present endogenous self-peptides, only loading of cognate peptides led to sustained T cell activation after Csk inhibition.

We speculated that the striking enhancement of weak agonist recognition is due to the strong inhibition of Csk and activation of Lck (threefold at 5 μM 3-IB-PP1). To test the extent of Csk inhibition

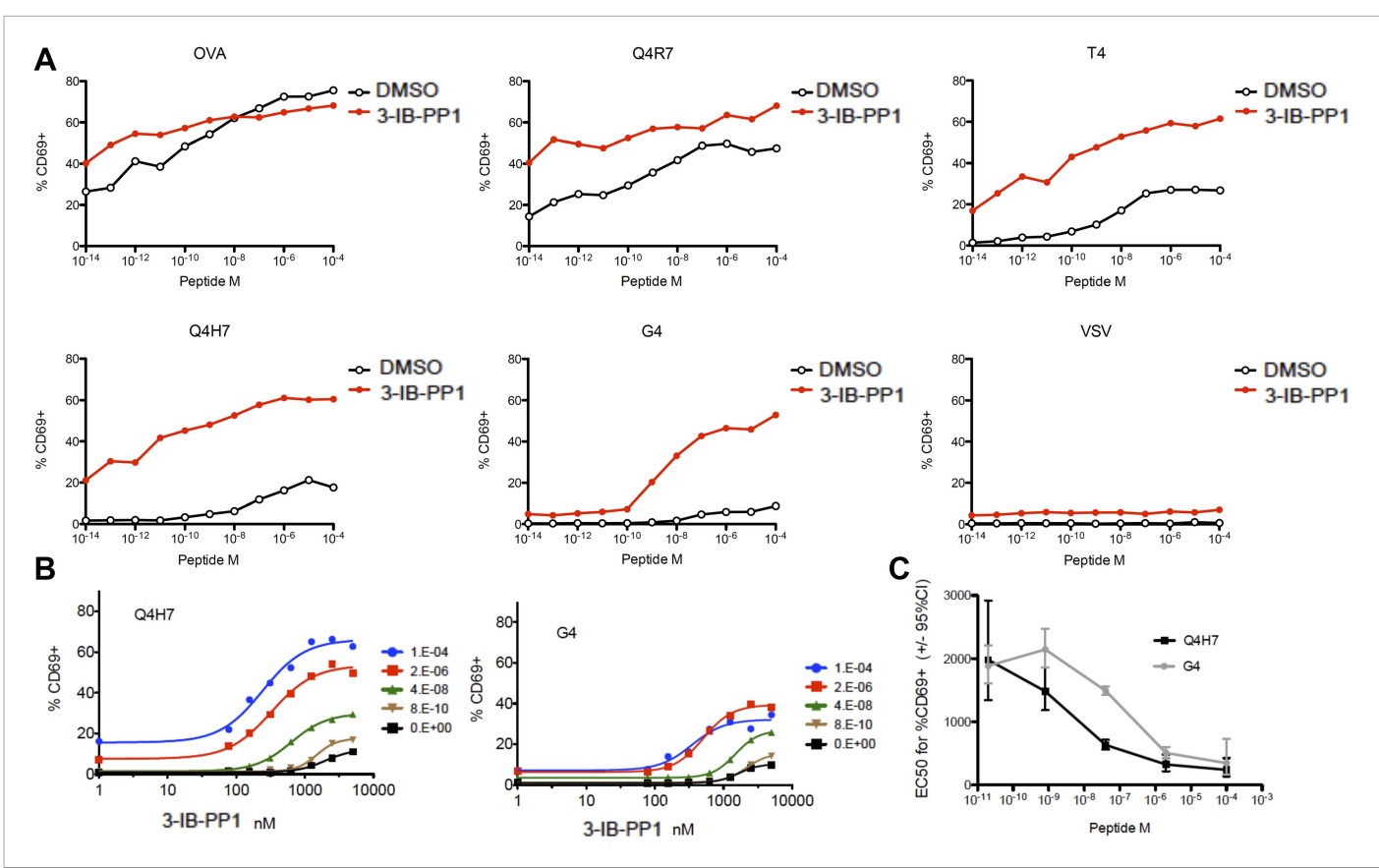

**Figure 4**. Csk inhibition potentiates response to weak agonists. (**A**) Csk[AS];OTI naive CD8[+] T cells were stimulated with MitoC-treated WT splenocytes, preloaded with indicated peptides, in presence of DMSO or 5 μM 3-IB-PP1 for 4 hr and CD69 upregulation was measured by flow cytometry. (**B**) Csk[AS];OTI naive CD8[+] T cells were stimulated as in **A** with varying doses of 3-IB-PP1 with dose–response fit. (**C**) EC50 of 3-IB-PP1 (nM) for % CD69 positive cells for data in **B**. Data are representative of two independent experiments.

The following figure supplement is available for figure 4:

**Figure supplement 1**. TCR Valpha2 downregulation in cells in *Figure 4A*.

necessary to boost weak agonist signaling, we titrated 3-IB-PP1 and assayed responses to weak agonists Q4H7 and G4 (*Figure 4B*). Surprisingly we observed dose-dependent enhancement of signaling at 100–1000 nM 3-IB-PP1, where increased Lck pY394 was weakly detectable and at most 50% above the basal level. Beyond this inhibitor dose, we observed a saturating effect, despite the lower overall plateau with lower peptide dose. Thus, relatively weak Lck hyperactivation was sufficient to enhance the recognition of weak agonists. The lower plateau suggests that there are Csk-independent mechanisms for sensing agonist dose. Remarkably, the EC50 concentration of Csk inhibition for CD69 upregulation was still dependent on agonist quality and dose (*Figure 4C*). At a lower agonist dose, the weaker peptide G4 required double the amount of 3-IB-PP1 than Q4H7. Similarly, the EC50 depended on peptide dose. These observations argue that Csk inhibition fine-tunes the response to agonist dose and affinity sensing in peripheral T cells.

## Discussion

Our results raise the question of how Csk activity controls SFK activity and specifically the TCR activation threshold. Since a substantial amount of active Lck is present in the basal state and does not change with TCR ligation, it has been hypothesized that the pre-existing pool of active Lck is responsible for the highly sensitive and rapid initiation of TCR signaling in response to ligand engagement (*Nika et al., 2010*). Hence, a likely explanation for our observations is that by controlling the amount of active Lck present, Csk regulates the responsiveness of the T cell to receptor stimulus.

Our data supports the model that the size of the active pool of Lck is especially critical for affinity discrimination of different agonistic peptides. Strong agonists have a long half-life of pMHC-TCR and have sufficient time to recruit active Lck and initiate downstream events, while weak agonists with short half-lives are most sensitive to the availability of active Lck before the pMHC-TCR complex falls apart. The different amounts of Lck loaded on the CD4 and CD8 co-receptors can explain the different half-life range of positively and negatively selecting ligands for CD4 and CD8 cells (*Stepanek et al., 2014*); however, this is a genetically established parameter. Our data supports the model that recruitment of active Lck is critical for affinity discrimination, but offers an alternative mechanism for its regulation. By controlling the fraction of active Lck, Csk can also modulate the activation threshold and affinity discrimination of T cells. Notably, since Csk localization and activity is dynamically controlled by phosphorylation-dependent recruitment to transmembrane and cytoplasmic adaptors, this could enable flexibility in affinity recognition in different settings. For example, Csk has been found to be differentially localized in naive and antigen-experienced T cells (*Borger et al., 2013*). Our results add to the emerging understanding of how signaling by agonists of dissimilar affinity are differentially regulated. Recently, the adaptor molecule Themis was characterized as critical for the suppression of weak agonists in thymocytes, while PTPN22 serves a similar role in in effector cells (*Fu et al., 2013*; *Salmond et al., 2014*). Here, we show that Csk, via its regulation of Lck, regulates naive T cell priming to agonists of different potency.

The ability to acutely titrate the activity of Csk, and consequently Lck, allowed us to assess how graded increases of Lck activity affects TCR signaling without concerns about basal level adaptation and long-term compensation that occurs when using genetic manipulation of Csk amounts or function. Surprisingly, even very small changes in Lck activity, caused by Csk$^{AS}$ inhibition at 200 nM 3-IB-PP1, led to relatively large pY394 changes in the setting of small changes in pY505. Since Y394 phosphorylation is the result of trans-autophosphorylation, it seems likely that a small reduction in pY505 induced by low levels of Csk inhibition, could lead to autoactivation of a large pool of Lck. Such a larger pool of active Lck could then show strong synergy with the TCR stimulus and is sufficient to markedly enhance the response to very weak agonists. The synergy we observed between TCR stimulus and Csk inhibition, strikingly, had its greatest impact on the activation of downstream signaling molecules. This is most likely due to positive feedback/signal amplification along the pathway. Eventually, the integration of stronger signaling over hours led to profound differences in the fraction of cells upregulating CD69 or proliferating. On the other hand, this minimal change in Lck activity highlights how tightly it must be regulated in both the basal and inducible states by kinases and phosphatases and their respective adaptors. It is possible that, although small changes in Lck activity greatly sensitize the TCR to weak agonists, such small changes do not as effectively engage negative feedback mechanisms.

Loss of Csk from the cell membrane, or biochemically defined lipid raft fractions has been proposed to lead to TCR signal initiation via activation of Lck (*Torgersen et al., 2001*; *Davidson et al., 2003*). This model is contradicted by the lack of appreciable change in Lck phosphorylation upon TCR

stimulation and by unaltered signaling in cells deficient in the lipid-raft-resident Csk adapter PAG (*Dobenecker et al., 2005*; *Xu et al., 2005*). Our data demonstrate that acute activation of Lck, via weak or strong Csk inhibition, does not lead to full TCR signaling. Strong Csk inhibition in mature T cells leads to small amounts of phosphorylation of downstream targets like ZAP-70 and PLC-γ1, but not ERK activation or long-term proliferation in the absence of TCR engagement. Thus, Lck activation alone is insufficient to trigger complete TCR signaling. However, when combined with cognate TCR ligands, very small amounts of Csk inhibition and resultant larger amounts of Lck activation, synergize to enhance early biochemical events and downstream activation. Therefore, it is possible that during physiological TCR stimulation, localization of only a small number of Csk molecules is altered and leads to the activation of a small fraction of Lck molecules that would be difficult to detect with current biochemical or imaging methods (*Paster et al., 2009*; *Nika et al., 2010*) but might be sufficient to influence downstream signaling.

Csk has also been proposed to regulate TCR signal downregulation by returning to the plasma membrane after signal initiation. It has been challenging to study this role using genetic ablation approaches since Csk deficiency affects TCR basal state signaling. Using acute inhibition of Csk concurrently with TCR stimulation, we were able to follow TCR signal downregulation in the absence of Csk activity and observed that it is delayed but not abolished. Our data suggest that other molecules are involved in signal termination more prominently than Csk.

New insights into how Csk is regulated will enable the development of strategies to manipulate Csk activity and hence the sensitivity of T cells to various activation stimuli. Such manipulation of Csk activity by small molecule inhibitors could be useful therapeutically, for example in pathologic settings where suppression (i.e., autoimmunity) or augmentation (i.e., vaccines or cancer) of T cell responses to antigen is desirable.

## Materials and methods

### Mice
Mice used for these studies were 6–12 weeks of age. All mice were housed in a specific pathogen-free facility at UCSF according to the University Animal Care Committee and National Institutes of Health (NIH) guidelines.

### Inhibitors
3-IB-PP1 has been described (*Schoenborn et al., 2011*).

### Antibodies and reagents
CD4 PerCP-Cy5.5 (550954), TCR Vα2 PE (553289) and Lck-pY505 (BD Biosciences 612390) are from BD Biosciences (San Jose, CA); p44/42 MAPK pThr202/Tyr204 (4377), Src 416 (2101), ZAP-70-pY319 (2701) are from Cell Signaling (Danvers, MA); LAT-pY132 (44-224), PLC-γ1-pY783(44-696G) and CFSE (C34554) are from Life Technologies (Life Technologies, Thermo Fisher Scientific, Waltham, MA); actin (Sigma Aldrich A2066; Sigma Aldrich, St. Louis, MO); GAPDH (Abcam ab8245; Abcam, Cambridge, MA); Goat anti-armenian hamster IgG(H+L) (127-005-160) and donkey anti-rabbit IgG Ab conjugated to APC (711-136-152) are from Jackson Immunoresearch (West Grove, PA); Horseradish peroxidase (HRP)-conjugated goat antibody α-rabbit IgG (H+L) (4050-05), α-mouse IgG (H+L) (1031-50), Alexa647-conjugated α-mouse IgG (H+L) (1010-31) are from Southern Biotech (Birmingham, AL); I-A(b) tetramers loaded with human class II-associated invariant chain peptide 103–117 or chicken OVA peptide 328–337 are from NIH tetramer core facility.

### Flow cytometry and data analyses
Stained cells were analyzed on a BD Fortessa (BD Biosciences; San Jose, CA). Data analysis was performed using FlowJo software (Treestar Incorporated; Ashland, OR) and GraphPad Prism (GraphPad Software, Inc.; La Jolla, CA) and ImageLab (BioRad Inc.; Hercules, CA).

### Cell stimulations and intracellular phosphoflow
Before stimulations, cells were serum-starved at 37°C for at least 20 min. Stimulations were performed in serum-free RPMI at 37°C. CD3ε crosslinking was induced by addition of anti-CD3ε followed by goat

anti-armenian hamster IgG(H+L) to a final concentration of 50 µg/ml. For bead-based stimulations, 4.5 µm styrene beads (Polyscience; Niles, IL) were coated overnight at 4°C or for 1 hr at room temperature with p-MHC tetramers or αCD3 (2c11) in PBS under continuous rotation at 4°C. Biotinylated p-MHC for CD8 stimulation were precoated with 5 µg/ml streptavidin. Beads were then saturated with 1% BSA in PBS under continuous rotation for 2 hr at room temperature, and washed with serum-free RPMI before use. Ice-cold rested cells and beads were span together and signaling was initiated by transfer to 37°C. Intracellular phospho-ERK was performed as previously described (*Schoenborn et al., 2011*).

## Immunoblotting
Immunoblotting was performed as previously described (*Tan et al., 2014*).

## Cell enrichments
Enriched populations of pan T cells were obtained by negative selection according to manufacturer's protocol (STEMCELL Technologies, 19751, 19851; STEMCELL Technologies; Vancouver, BC, Canada). Enriched populations of CD4+ or CD8+ T cells were obtained by negative selection according to manufacturer's protocol (STEMCELL Technologies 19852) or with an in-house procedure as follows: The following biotinylated antibodies were combined and dialyzed in 1× PBS with Slide-a-lyzer 10,000 Molecular Weight cutoff dialysis cassette (Thermo catalog# 66810; Thermo Fisher Scientific, Waltham, MA), then filter sterilized: biotin anti-CD8a (clone 53-6.7) or biotin anti-CD4 (clone GK1.5), biotin anti-CD11b (clone M1/70), biotin anti-CD11c (clone N418), biotin anti-CD19 (clone 1D3) from were Tonbo Biosciences (San Diego, CA); biotin anti-CD24 (clone M1/69), biotin anti-CD45R (B220) (clone RA3-6B2), biotin anti-CD49b (clone DX5), biotin anti-TER119 (clone TER-119) were from Biolegend (La Jolla, CA). ACK-lysed splenocytes and lymphocytes were incubated at room temperature with the antibody cocktail for 10 min in PBS with 1% FBS, 2 mM EDTA and 5% normal rat serum at $10^8$ cells/ml. Cells were then washed once with PBS with 1% FBS, 2 mM EDTA, resuspended at 0.85 ml per $10^8$ cells, and mixed with 0.15 ml anti-biotin MACSiBead per $10^8$ cells (Miltenyi Biotec; San Diego, CA). After 5 min at room temperature, cell-bead mix was placed in separation magnet for 5 min and the unbound cells were collected.

## Cell proliferation
Purified CD4+ or CD8+ T cells were resuspended in PBS, labeled for 4 min at room temperature with 2 µM CFSE, and quenched with 100% FBS. Labeled CD4+ T cells were stimulated by plate-coated OVA-MHC or α-CD3 (2c11) for 72 hr. At indicated timepoint, DAPI were analyzed for CFSE dilution.

## CD69 upregulation
Purified CD8+ OTI T cells were stimulated for indicated time by Mitomycin C-treated Calpha−/− or BoyJ splenocytes, preloaded for 1 hr with indicated peptide and washed of excess peptide. At indicated time, cells were fixed and stained for CD69, TCRValpha2 and CD45.1, and analyzed by flow cytometry.

## Acknowledgements
We thank Al Roque for assisting with animal husbandry. This work was supported in part by NIH grants PO1 AI091580 (AW), R01 GM107671 (KMS), NSF Pre-Doctoral Fellowship (FP), Cancer Research Institute Postdoctoral Fellowship (BNM) and the Robertson Foundation/Cancer Research Institute Postdoctoral Fellowship (AC).

## Additional information

### Funding

| Funder | Grant reference | Author |
| --- | --- | --- |
| National Institutes of Health (NIH) | PO1 AI091580 | Boryana N Manz, Ying Xim Tan, Adam H Courtney, Arthur Weiss |

| Funder | Grant reference | Author |
|---|---|---|
| National Science Foundation (NSF) | Pre-doctoral fellowship | Florentine Rutaganira |
| Cancer Research Institute (CRI) | Postdoctoral Fellowship | Boryana N Manz, Adam H Courtney |
| National Institutes of Health (NIH) | R01 GM107671 | Florentine Rutaganira, Kevan M Shokat |

The funders had no role in study design, data collection and interpretation, or the decision to submit the work for publication.

## Author contributions

BNM, YXT, Conception and design, Acquisition of data, Analysis and interpretation of data, Drafting or revising the article; AHC, Acquisition of data, Analysis and interpretation of data, Drafting or revising the article; FR, Conception and design, Drafting or revising the article, Contributed unpublished essential data or reagents; EP, KMS, Drafting or revising the article, Contributed unpublished essential data or reagents; AW, Conception and design, Analysis and interpretation of data, Drafting or revising the article

## Ethics

Animal experimentation: This study was performed in strict accordance with the recommendations in the Guide for the Care and Use of Laboratory Animals of the National Institutes of Health. All of the animals were handled according to approved institutional animal care and use committee (IACUC) protocols (#AN086836, AN107127) of the University of California, San Francisco.

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
