## [Decision Letter]

Thank you for submitting your work entitled “Small molecule inhibition of Csk alters affinity recognition by T cells” for peer review at *eLife*. Your submission has been favorably evaluated by Tadatsugu Taniguchi (Senior editor) and three reviewers, one of whom is a member of our Board of Reviewing Editors.

The reviewers have discussed the reviews with one another and the Reviewing editor has drafted this decision to help you prepare a revised submission.

Summary:

The authors use the Csk-AS model to investigate the impact of acute Csk inhibition on both CD3 Ab and MHC-peptide-induced biochemical signaling pathways and downstream effects on activation marker expression, TCR downregulation and proliferation. The major conclusions that Csk 1) regulates the basal activity of Lck (and Fyn) and 2) influences the extent of T cell activation in response to low-dose / low affinity Ags are supported by the data.

Furthermore, Csk inhibition only weakly prolongs the duration of TCR-induced signaling – suggesting that other inhibitory pathways are required for signal termination. Overall, this elegant study establishes that Csk controls the size of the active pool of Lck and thus contributes to set the TCR signaling threshold and thus enforces TCR affinity discrimination in peripheral T cells.

Essential revisions:

1) Using CskAS expressing OT1 T cells, the authors made the puzzling finding that the phosphorylation of the inhibitory tail pY505, the direct target of Csk, did not change in response to 3-IB-PP1 treatment. In contrast, a 3-4 fold increase in Y394. This observation is inconsistent with most present model(s) of Lck regulation. How do the authors account for this intriguing observation?

2) Results, first paragraph. The recent work of Ballek et al. (doi:10.1038/icb.2014.100) suggests that the pool of active Lck is smaller than reported and that stimulation of T cells by physiological TCR and CD4 ligation does increase the number of active Lck molecules. This work needs to be discussed.

3) In Figure 3 – these are (to me at least) a little confusing. Any effect of Csk inhibition on levels of Lck pY394 would be secondary to direct effects on pY505. It is possible that to see changes in pY505 following Csk inhibition requires a dephosphorylation event by CD45, but it seems a bit odd that large changes in pY394, as seen with high doses of 3-IB-PP1, should be apparent without any change in pY505? Given the results in Figures 1 and 3 appear to give quite different results as regards the effects of the drug on basal pY505 in CD4 (Figure 1) and OT-1 (Figure 3) cells, this should be discussed. Is this a “real” effect – i.e., is pY505 regulated differently is CD4 and CD8 T cells?

---

## [Author Response]

1) Using CskAS expressing OT1 T cells, the authors made the puzzling finding that the phosphorylation of the inhibitory tail pY505, the direct target of Csk, did not change in response to 3-IB-PP1 treatment. In contrast, a 3-4 fold increase in Y394. This observation is inconsistent with most present model(s) of Lck regulation. How do the authors account for this intriguing observation?

The reviewers point out the puzzling observation that phosphorylation of the direct target of Csk, Lck Y505, does not change much upon Csk inhibition, whereas phosphorylation of the activating site Y394 on Lck goes up dramatically.

Different cell types from Csk^AS^ mice have been examined and they all respond with Src family kinase (SFK) activation after Csk inhibition, however the phosphorylation status of the inhibitory tyrosines differs substantially. In thymocytes ([29]
doi:10.1038/ni.2772) only partial loss is observed at high drug doses, even after prolonged treatment, while in bone marrow derived macrophages, the inhibitory tyrosines of the SFKs are rapidly and almost completely de-phosphorylated (Freedman, et al., manuscript in revision for *eLife* #02-06-2015). Moreover, similar more complete dephosphorylation of the inhibitory site is observed in B cells (unpublished). Likewise, in Jurkat T cells expressing a membrane targeted CskAS construct that functions a dominant inhibitory allele, inhibition of Csk led to near complete dephosphorylation of the Lck inhibitory site (Schoenborn, et al., Sci. Signal., 2011). We conclude that the SFKs are differentially regulated in different cell types, perhaps via differential sequestration from CD45, the contribution of other phosphatases such as CD148, or pools that are prone to autophosphorylation of the activating site pY394. To address the difference between CD4^+^ and CD8^+^ T cells, we immunoblotted lysates from equivalently stimulated cells side by side (new Figure 3—figure supplement 1). The main observation is that both respond similarly to Csk inhibition, with a smaller change in pY505 than in pY394. CD8^+^ T cells have ∼ 15% more Lck and higher pY505 level at basal state, which may account for the higher residual pY505 level in CD8^+^ T cells, especially at low levels of Csk inhibition. Moreover, it is possible that small reductions in pY505 allow for initial increases in pY394 but the small increase in pY394 could lead to trans-phosphorylation of other Lck molecules at pY394, i.e., an auto-catalytic function.

We would like to emphasize that the effect on affinity recognition in CD8^+^ T cells is observed at very low doses of the drug, where both changes in Lck pY505 and pY394 are very subtle. Most of the biochemical work with CD8^+^ T cells was also performed at lower drug doses than used for CD4^+^ T cells; so, one cannot directly compare the data presented in the paper from the two different cell types. We believe the new supplemental figure clarifies the reviewer's concern.

*2) Results, first paragraph. The recent work of Ballek et al. (**doi:10.1038/icb.2014.100**) suggests that the pool of active Lck is smaller than reported and that stimulation of T cells by physiological TCR and CD4 ligation does increase the number of active Lck molecules. This work needs to be discussed.*

The reviewers point out a recent estimate of active Lck in T cells (Ballek et al.) that is much lower than a previous one (Nika et al.) with which our data is consistent.

Our study was not designed to answer the question of how much Lck is active in the basal state, which was the focus of both of the above studies, and they approached it with different experimental techniques. By acute Csk inhibition, we were able to activate significantly more Lck molecules and based on the plateau level, we inferred an estimate of the basally active fraction. Although it is consistent with Nika et al.'s results, it is possible that there remains a pool of Lck that cannot be activated acutely by Csk inhibition, due to sequestration away from Csk, proximity to CD45 or another mechanism. Therefore our approach is not suited to quantitatively answer this question. This distinction has been noted in the revised text.

*3) In*
Figure 3
*– these are (to me at least) a little confusing. Any effect of Csk inhibition on levels of Lck pY394 would be secondary to direct effects on pY505. It is possible that to see changes in pY505 following Csk inhibition requires a dephosphorylation event by CD45, but it seems a bit odd that large changes in pY394, as seen with high doses of 3-IB-PP1, should be apparent without any change in pY505? Given the results in*
Figures 1 and 3
*appear to give quite different results as regards the effects of the drug on basal pY505 in CD4 (*Figure 1*) and OT-1 (*Figure 3*) cells, this should be discussed*. *Is this a “real” effect – i.e., is pY505 regulated differently is CD4 and CD8 T cells?*

The reviewers point out the surprising activation of pY394 with very subtle changes in pY505, which was addressed in point 1, and the apparent discrepancy between CD4^+^ and CD8^+^ T cells.

To quantitatively address any differences between CD4^+^ and CD8^+^ T cells, we compared them side by side, as seen in the new Figure 3—figure supplement 1. They respond fairly similarly to each other following Csk inhibition, with CD8^+^ T cells having slightly more Lck and a larger fraction of it is phosphorylated at Y505. It is important to note that most of the data with CD8^+^ T cells was performed with much lower doses of the Csk inhibitor where changes at the Lck phosphorylation sites were relatively small. On the other hand, the CD4^+^ T cell experiments used high drug doses (5 or 10 µM) where greater change was observed.

The mechanism behind the relatively small changes in pY505 and larger changes in pY394 is unknown. As mentioned in our response to point 1, we hypothesize that the trans-autocatalytic nature of Lck may account for dramatic changes in the activation site. This would allow a small number of Lck molecules that are initially activated through Csk inhibition to rapidly phosphorylate the active site of other Lck molecules and amplify the overall level of activation. This explanation has been added to the text.